# Long-range focusing of magnetic bound states in superconducting lanthanum

Howon Kim [1✉], Levente Rózsa [1,3], Dominik Schreyer[1], Eszter Simon [2,4] & Roland Wiesendanger [1✉]

Quantum mechanical systems with long-range interactions between quasiparticles provide a promising platform for coherent quantum information technology. Superconductors are a natural choice for solid-state based quantum devices, while magnetic impurities inside superconductors give rise to quasiparticle excitations of broken Cooper pairs that provide characteristic information about the host superconductor. Here, we reveal that magnetic impurities embedded below a superconducting La(0001) surface interact via quasiparticles extending to very large distances, up to several tens of nanometers. Using low-temperature scanning probe techniques, we observe the corresponding anisotropic and giant oscillations in the LDOS. Theoretical calculations indicate that the quasi-two-dimensional surface states with their strongly anisotropic Fermi surface play a crucial role for the focusing and long-range extension of the magnetic bound states. The quasiparticle focusing mechanism should facilitate the design of versatile magnetic structures with tunable and directed magnetic interactions over large distances, thereby paving the way toward the design of low-dimensional magnet–superconductor hybrid systems exhibiting topologically non-trivial quantum states as possible elements of quantum computation schemes based on Majorana quasiparticles.

[1] Department of Physics, University of Hamburg, D-20355 Hamburg, Germany. [2] Department of Theoretical Physics, Budapest University of Technology and Economics, Budafoki út 8, H-1111 Budapest, Hungary. [3] Present address: Department of Physics, University of Konstanz, D-78457 Konstanz, Germany. [4] Present address: Department Chemie, Physikalische Chemie, Ludwig-Maximilians-Universität München, Butenandtstr. 5-13, D-81377 München, Germany. ✉email: hkim@physnet.uni-hamburg.de; wiesendanger@physnet.uni-hamburg.de

The coherent propagation and scattering of quasiparticle excitations are fundamental for understanding various physical and chemical phenomena in condensed matter systems. In metals, the electrons at the Fermi surface (FS) are prominent in determining thermodynamic properties, and in explaining magnetic ordering, charge-density waves, and super-conductivity[1]. The FS is defined in the reciprocal space of crystals, but its influence is also manifested in real space in the vicinity of impurities via screening of the impurity potential, resulting in a modulation of the electron density, also known as Friedel oscillations[2]. Anisotropic FSs effectively focus and guide the scattered quasiparticles along specific directions, enabling the characterization of nonmagnetic and magnetic impurities at large distances[3,4].

In a conventional superconductor, magnetic impurities locally break Cooper pairs and generate quasiparticle excitations inside the superconducting gap, known as Yu–Shiba–Rusinov (YSR) bound states[5–7]. Arranging the magnetic impurities into low-dimensional arrays has recently attracted great interest in the pursuit of Majorana bound states (MBS), which are promising as a key element in topological quantum computation[8–12]. The long-range and anisotropic interaction of the impurities reflected in the spatial extent of the YSR states allows the tailoring of MBSs in low-dimensional magnetic systems on superconductors.

Scanning tunneling microscopy (STM) and spectroscopy (STS) measurements provide access to the YSR states at the atomic level with high spatial and energy resolution. Using these techniques, the spatial extension of YSR states for most magnetic impurities on surfaces of three-dimensional (3D) superconductors was found to be limited to the range of a few atomic lattice constants[12–16]. A significantly longer extension of a few nanometers was observed for Mn atoms on Pb(111), and has been attributed to the projection of the bulk FS onto this crystallographic direction[17]. Recently, Ménard et al. reported a slow decay of YSR states for subsurface Fe impurities inside the layered superconductor 2H–NbSe$_2$, observable at distances up to several nanometers[18]. This was attributed to the essentially two-dimensional (2D) electronic structure caused by the weak hybridization between the layers, and it was argued that layered materials are preferable for studying long-range interactions between YSR states. The sizeable interaction between YSR states around magnetic molecules on NbSe$_2$ was indeed demonstrated by Kezilebieke et al.[19].

Here, we demonstrate that YSR bound states with anisotropic and extremely long-range extension, reaching up to several tens of nanometers, can be observed on the (0001) surface of the 3D elemental superconductor La using low-temperature STM/STS. This is remarkably longer than what has been reported based on previous STM/STS investigations on any 3D or even quasi-2D superconductor[12–14,16,18]. Furthermore, we confirm the presence of the interaction between impurities at distances of several nanometers. Corroborated by first-principles and model calculations, we show that this extraordinary long-range modulation of the local density of states (LDOS) due to YSR impurities provides direct information on the anisotropic FS formed by the quasi-2D surface states through the quasiparticle-focusing effect.

## Results

**Embedded YSR impurities below the La(0001) surface**. We have studied bulk-like α-lanthanum films epitaxially grown on a Re(0001) single crystal [see "Methods" and Supplementary Note 1 and Supplementary Fig. 1]. STM/STS measurements were carried out at $T = 1.65$ K, which is well below the superconducting transition temperature of La films, $T_c \sim 5$–6 K[20]. The magnetic impurities were introduced during the La deposition, being

contaminants in the 99.9+ % pure La material used for the experiments [see "Methods"].

The magnetic impurities are clearly identifiable in differential tunneling conductance ($dI/dV$) maps taken above the atomically flat La(0001) surface at a bias voltage corresponding to states inside the superconducting gap of La, as shown in Fig. 1a. Since there are no quasiparticle excitations in the bulk superconductor at this energy, the local modulation of the LDOS can solely be attributed to the YSR bound states formed by the impurities, which are embedded below the La(0001) surface [see Supplementary Note 2 and Supplementary Fig. 2]. The various shapes of the modulations (YSR1, YSR2, YSR3, and YSR4 in Fig. 1a, b) presumably originate from the various depths of the magnetic impurities [see Supplementary Note 3 and Supplementary Fig. 3]. However, it is also possible that different magnetic elements being present as natural impurities even in high-purity La material may contribute to the different spatial and spectroscopic signatures of the observed YSR impurities. In the following, we mainly focus on YSR1-type impurities, characterized by the highest modulation intensity.

Figure 1b, c shows zoomed-in $dI/dV$ maps for the YSR1 impurity obtained at positive and negative bias voltages, or electron- and hole-like bound states, respectively. The spatial extension of the YSR bound state resembles a star-shaped pattern, with six beams protruding along the directions parallel to the reciprocal lattice vectors (see the inset of Fig. 1a), and an oscillatory modulation characteristic of YSR states[6,7]. The oscillations are observable up to ~30 nm away from the impurity along the beams, and up to ~10 nm distance along the directions halfway between the beams, both being considerably longer than in previously reported STM/STS studies of YSR bound states[12–14,16,18].

**Spectroscopic signature of spatially extended YSR states**. In Fig. 1d, the YSR quasiparticle excitations show up as a pair of resonances in the $dI/dV$ spectra inside the superconducting gap of La ($\Delta_{\text{La,surface}} \sim 1$ meV). Taking into account the convolution of the Bardeen–Cooper–Schrieffer-type DOS of the sample and the La-coated superconducting tip in the measured tunneling spectrum[21], the binding energies of the YSR states are found at $E_{\text{YSR}} = \pm 0.55$ meV (red and blue arrows). Their symmetric position with respect to the Fermi level reflects the particle–hole symmetry of the superconductor.

The spatial extension of the YSR states shown in Fig. 1b, c has been quantitatively analyzed by taking differential tunneling conductance profiles (Fig. 1e) and spectroscopic maps (Fig. 1f) along lines crossing the center of the magnetic impurity. The period of the oscillations, corresponding to half of the Fermi wavelength ($\lambda_F/2$), is found to be ~0.99 nm along the beams (**Q2** direction in Fig. 1e) and ~0.90 nm along the direction halfway between the beams (**Q1** direction), respectively, for both positive and negative bias voltages. The phase shift between the spectroscopic maps at $\pm E_{\text{YSR}}$, determined by the binding energy[17,18], is clearly conserved as far as 30 nm away from the impurity along the **Q2** direction, confirming that even at this distant point, the LDOS modulation can be attributed to the coherent particle–hole symmetric YSR states.

**Quasi-2D surface electronic structure of La(0001)**. To address the origin of the anisotropic long-range extension of the YSR states, we turn our attention toward the surface electronic structure of La(0001). The $dI/dV$ spectrum averaged over a flat terrace shows a sharp resonance peak at $E = +110$ meV, which reflects the band edge of the surface state of La(0001) as shown in the left panel of Fig. 2a [Supplementary Note 1][22]. The

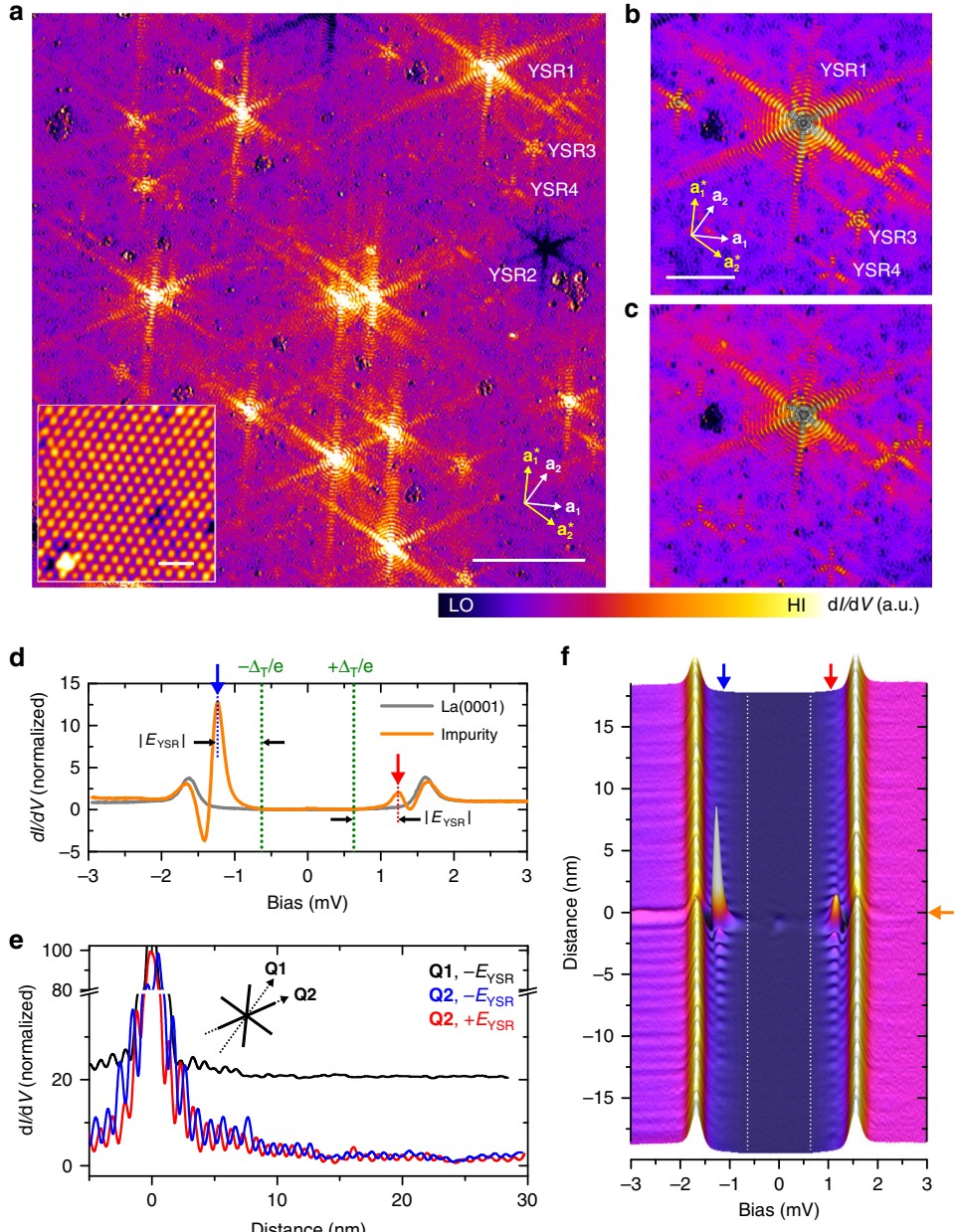

**Fig. 1 Anisotropic long-range modulation of Yu–Shiba–Rusinov (YSR)- bound states around magnetic impurities on a superconducting La(0001) surface. a** Differential tunneling conductance ($dI/dV$) map showing star-shaped LDOS modulations of the YSR bound states around magnetic atoms. Tunneling parameters: $I_T = 1.0$ nA, $V_S = -1.2$ mV, $150 \times 150$ nm$^2$, scale bar: 30 nm. (Inset: Atomically resolved STM constant-current image. Scale bar: 1.0 nm). The white arrows, $\mathbf{a_1}(\mathbf{a_2})$ and the yellow arrows, $\mathbf{a_1}^*(\mathbf{a_2}^*)$ denote the lattice vector directions on the La(0001) surface extracted from the STM image and the corresponding reciprocal lattice vectors, respectively. **b, c** Zoomed-in $dI/dV$ maps for an isolated magnetic atom YSR1 (white arrow in **a**) showing oscillatory beam-like extensions of the YSR bound state at **b** $V_S = -1.2$ mV and **c** $V_S = +1.2$ mV, $60 \times 60$ nm$^2$, $I_T = 1.0$ nA, scale bar: 15 nm. **d** Tunneling spectra measured on a single magnetic impurity (orange) and on the La(0001) surface (gray). A superconducting La-coated PtIr tip with a gap size of $\Delta_T = 0.65$ meV was used for taking STM images and differential tunneling conductance maps. A pair of the YSR bound states is indicated by red and blue arrows at $\pm|\Delta_{tip} + E_{YSR}|$ with $E_{YSR} = 0.55$ meV (black arrows). **e** Differential tunneling conductance profiles taken across the center of a magnetic atom along two different directions, **Q1** and **Q2**, as depicted in the inset. The profile along **Q1** is plotted with an offset for clarity. **f** One-dimensional tunneling spectroscopic map across the YSR1 impurity along the **Q2** direction.

atomic defects close to the surface create oscillatory quasi-particle interference (QPI) patterns in the measured $dI/dV$ maps [see Supplementary Note 4 and Supplementary Fig. 4]. The Fourier transform (FT) of an STM image obtained at a bias voltage just outside the superconducting gap (Fig. 2c) provides direct information about the FS[23], possessing a hexagonal shape

with flat parts along the **Q2** direction and rounded vertices along the **Q1** direction. By quantitatively analyzing the FT maps as a function of bias voltage, the dispersion relation along the **Q1** and **Q2** directions can be derived from the varying size of the hexagon (right panel of Fig. 2a [see Supplementary Note 4]).

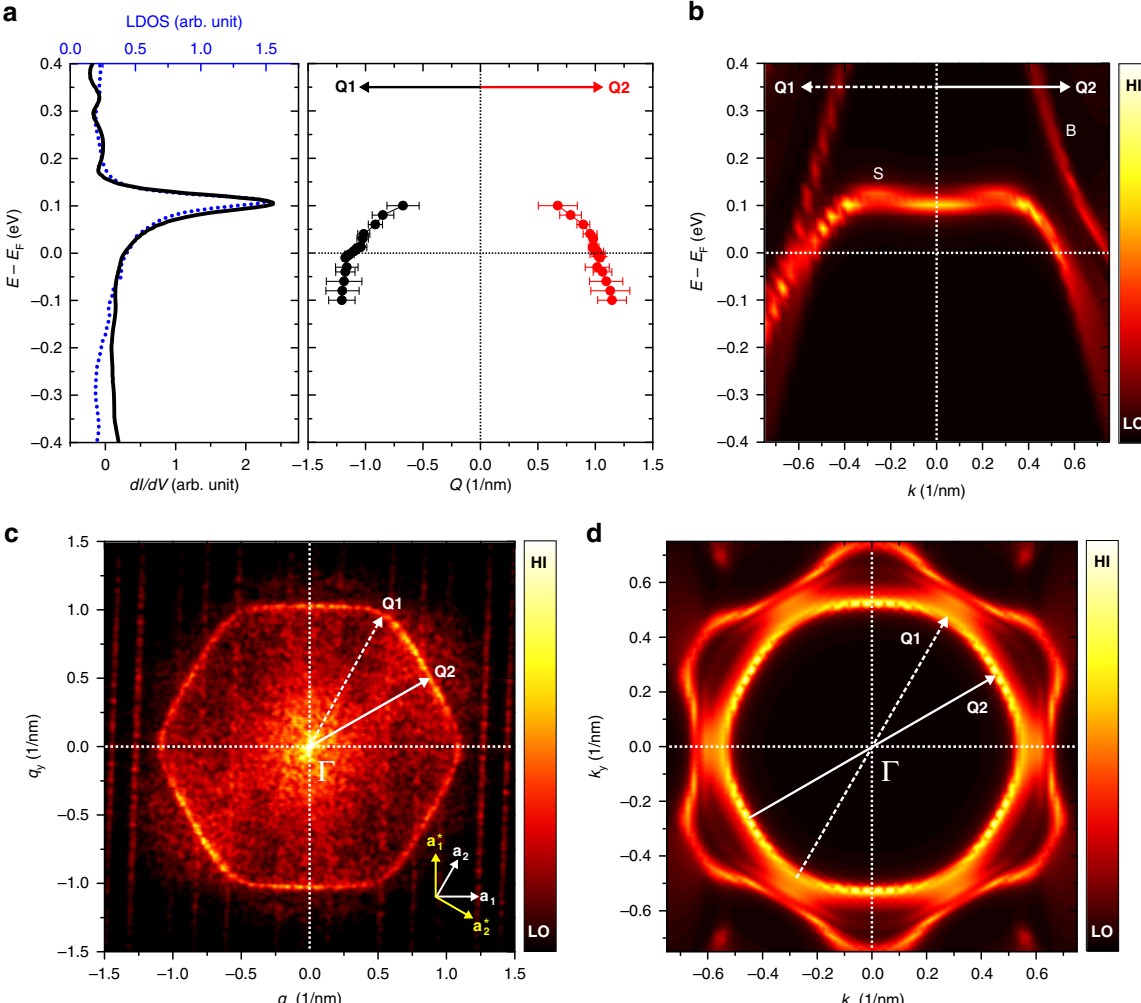

**Fig. 2 Quasi-two-dimensional surface electronic structure of La(0001). a** (left) Differential tunneling conductance spectrum in a wide energy range on a clean La(0001) surface (black solid line) and calculated LDOS in the vacuum 3 Å above the surface (blue dotted line). The pronounced peak around $E = 0.1$ meV can be attributed to a d-type surface state. (right) Extracted dispersion relation from the FT of the bias-dependent $dI/dV$ maps (see [Supplementary Note 4 and Supplementary Fig. 4]) along the **Q1** and **Q2** directions. The error bars correspond to the full width at half maximum of the Gaussian fitting for the peak in the line section of FT maps. **b** Calculated Bloch spectral function (BSF) for the surface atomic layer of La(0001) along the **Q1** and **Q2** directions. **c** The 2D FT of the $dI/dV$ map at $V_S = +3.0$ mV showing a hexagonal scattering pattern. **d** BSF at the Fermi energy for the surface atomic layer, highlighting the FS formed by the surface bands. The white arrows present the corresponding quasiparticle scattering processes in (**c**).

Since both the quasiparticle scattering at the Fermi energy and the YSR states find their origin in the FS, it is reasonable to consider the analogies between them. They share the same period of oscillation along the two directions **Q1** and **Q2**. Due to the quasiparticle-focusing effect, both types of oscillations are extended to a longer range along the flat directions of the FS than along the vertices, as can be seen when comparing Fig. 2 with Fig. 1.

The experimental QPI results are remarkably well reproduced by the Bloch spectral function (BSF) calculated based on the screened Korringa–Kohn–Rostoker method within density functional theory, providing information on the electronic band structure [see "Methods"]. Two high-intensity spectral features are observed close to the Γ point (Fig. 2b): the bulk band (B) is present in all layers, while the surface band (S) appears only in the top few atomic layers, with a flat dispersion around $E = 0.1$ eV, characteristic of localized d-type surface states. The BSF at the Fermi energy in the surface Brillouin zone is displayed in Fig. 2d. Since the momentum transfer vectors **Q1** and **Q2** are twice as

long as the Fermi wave vectors along the relevant directions, it can be concluded that the hexagonal QPI pattern in Fig. 2c arises from intraband scattering of quasiparticles originating from the surface band (inner hexagon in Fig. 2d). The QPI pattern and the YSR states appear to be insensitive to the bulk band (outer hexagon present in the BSF calculations in Fig. 2d), which is rotated by 30° with respect to the surface band. This implies that the surface band of La(0001) plays a major role in the quasiparticle scattering at the surface, while the contribution of the bulk band is weak or lacking. Note that similarly to the present case, only one of the two FSs contributes to the observed YSR states in Pb(111)[17] as well as in NbSe$_2$[18].

**Quasiparticle focusing of YSR states via the FS.** The QPI patterns and the first-principles calculations thereby reveal two major contributions to the anisotropic long-range extension of the YSR states. The first is the quasi-2D electronic structure, because primarily electrons in the surface band contribute to the scattering processes. The second is the quasiparticle-focusing

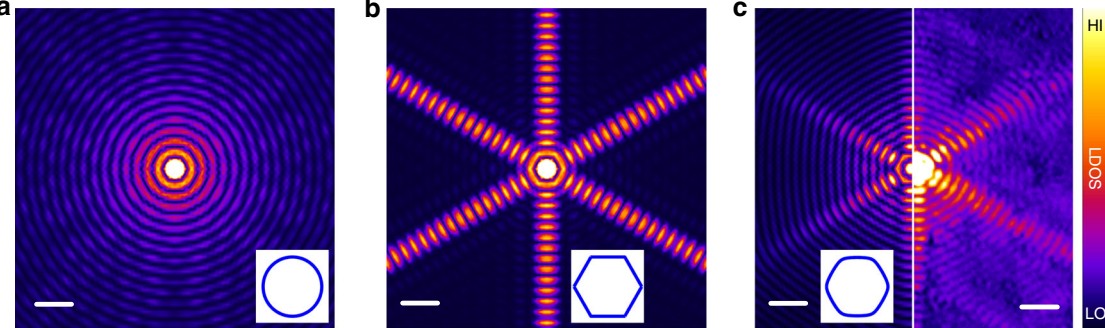

**Fig. 3 Influence of the shape of the Fermi surface on the focusing of Yu–Shiba–Rusinov bound states. a, b** Numerically calculated LDOS maps around a single magnetic impurity for the hole-like YSR bound state considering **a** a circular and **b** a hexagonal Fermi surface (FS). **c** Direct comparison between the calculated LDOS with the realistic FS of La(0001) (left) and the experimentally obtained $dI/dV$ map at $E = -E_{YSR}$ (right). Scale bar: 10 atomic sites (simulations), 4.0 nm (experiment). For details of the model and the calculation parameters, see Methods, Supplementary Notes 5 and 6, Supplementary Figs. 5 and 6. The corresponding shapes of the FS are depicted in the inset for each calculated image. The intensity scale of all calculated LDOS maps is the same, to be able to compare them directly with each other.

effect, causing beam-like extensions perpendicular to the flat parts of the anisotropic FS. In order to quantify the role of the shape of the FS on the spatial extension, we performed 2D atomistic model calculations parameterized by the first-principles results to determine the energy and the LDOS modulation of the YSR states [see Methods]. As shown in Fig. 3a, an ideal circular FS leads to an isotropic decay of the YSR intensity, following a $1/r$ power law at intermediate distances where $r$ is the distance from the impurity[17,18]. For the hexagonal FS in Fig. 3b, the LDOS along the beam directions perpendicular to the flat sides of the FS decays slower than $1/r$, effectively reducing the dimensionality due to the strong focusing effect [see Supplementary Note 5, Supplementary Fig. 5, and Supplementary Movie 1][3,24]. By considering the realistic FS of La(0001) obtained from the first-principles calculations, which is slightly rounded from the ideal hexagon as shown in Fig. 2c, d, our numerical results (Fig. 3c, left) are consistent with the experimentally obtained $dI/dV$ map for the YSR1 impurity (Fig. 3c, right) with regard to the shape and the range of the extension, as well as the oscillation period.

It is worth mentioning that the focusing due to the anisotropic FS and the dimensionality of the system cannot rigorously be distinguished, because they are essentially different manifestations of the same mechanism. For example, an isotropic two-dimensional FS (circle) corresponds to a cylindrical three-dimensional FS, which is strongly anisotropic since the quasiparticles cannot propagate along the axis of the cylinder. Incidentally, the innermost sheet of the FS of bulk La is almost perfectly cylindrical according to our calculations [see Supplementary Fig. 6]. Conversely, the contribution of the anisotropic FS may be understood as an effectively reduced dimensionality [see Supplementary Note 5 and Supplementary Fig. 5]. However, we find that a circular two-dimensional or cylindrical three-dimensional FS could not fully account for the long-range extension, since they would lead to a characteristic $1/r$ decay that is faster than the spatial decay observed in the experiments ($\sim 1/r^{0.92}$).

**Interacting YSR impurities at large distances**. Finally, we demonstrate that the YSR impurities on La(0001) exhibit a significant long-range coupling between them. The interaction between YSR impurities results in a variation of $E_{YSR}$ compared to the case of a single impurity[19,25–28]. Figure 4a presents four YSR1-type impurities: two of them (A, B) at about 3.6 nm distance roughly along the **Q2** direction, and two isolated ones

(I1, I2) being significantly further away. Interestingly, due to the interference between the YSR states of the impurities A and B, the LDOS shows a twin-star pattern with inhomogeneous modulations around the paired YSR1-type impurities. As shown in Fig. 4b, the binding energy $E_{YSR} = 0.56$ meV for the isolated I1 and I2 impurities is consistent with the one for the YSR1 impurity in Fig. 1d. However, on the A and B impurities, a sizeable shift of around 90 μeV is observed, being identical for the two atoms. This clearly indicates that the extended YSR bound states give rise to mutual long-range interactions between the magnetic impurities.

## Discussion

In conclusion, we have demonstrated how the anisotropic FS formed by the surface band on La(0001) causes magnetic bound states in the superconducting phase to extend coherently over tens of nanometers along preferential directions, the extension being about an order of magnitude longer than what has previously been reported for other bulk elemental superconductors. The results confirm that the slow power-law decay and long-range extension of the YSR bound states, which may be interpreted as an effective reduction in dimensionality, can be observed not only in layered systems, but also in elemental bulk superconductors exhibiting distorted FSs with quasi-2D bands. Moreover, the large spatial extent of YSR states results in a coupling between the magnetic impurities at remarkably long distances. These findings could potentially be combined with the remote control of the magnetic exchange interactions between localized spins already demonstrated in STM/STS experiments[29,30], the range and direction of which is similarly determined by the FS via the Ruderman–Kittel–Kasuya–Yosida mechanism. Engineering the band structure of YSR states[31,32] together with the configuration of arrays of magnetic impurities will facilitate the design of artificially built low-dimensional magnet–superconductor hybrid systems exhibiting topologically nontrivial quantum states[12].

## Methods

**Preparation of the sample and tip**. The rhenium single crystal was prepared by repeated cycles of $O_2$ annealing at 1400 K followed by flashing at 1800 K to obtain an atomically flat Re(0001) surface[33]. The over 50-nm-thick lanthanum films were prepared in situ by electron beam evaporation of pure La pieces (99.9+%, MaTeck, Germany) in a molybdenum crucible at room temperature onto a clean Re(0001) surface, followed by annealing at 900 K for 10 min[20]. The films were thicker than the coherence length of bulk La ($\xi = 36.3$ nm)[34] in order to suppress the proximity effect from the Re substrate [see Supplementary Note 1 and Supplementary Fig. 1].

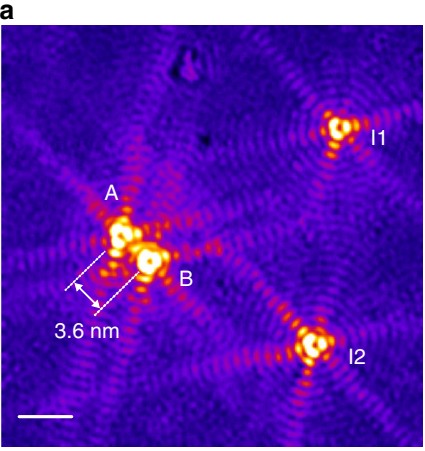

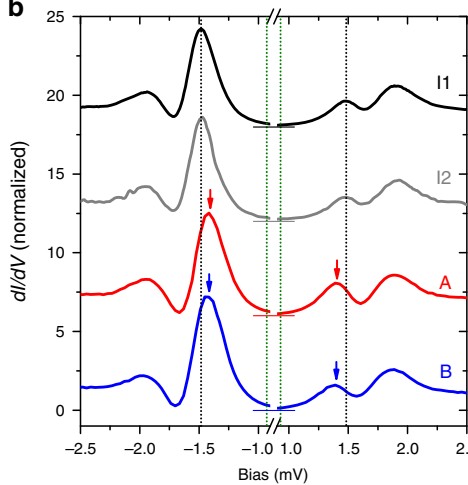

**Fig. 4 Long-range coupling between YSR impurities on La(0001). a** Differential tunneling conductance map showing spatial modulations of the magnetic bound states around four YSR1 impurities. Two impurities (A and B) are separated by a distance of 3.6 nm, comparable to 10 atomic lattice constants on the La(0001) surface, while the other two (I1 and I2) are spatially isolated. $I_T = 1.0$ nA, $V_S = 1.2$ mV, $T = 1.67$ K. Scale bar: 5.0 nm. **b** Tunneling spectra obtained at the centers of the four magnetic impurities. Black dotted lines denote the energies of $\pm|\Delta_T + E_{YSR}|$ for the isolated YSR1 impurities (I1 and I2) with $E_{YSR} = 0.56$ meV. The edge of the superconducting gap ($\Delta_T = 0.93$ meV) of the La-coated tip is depicted by green dotted lines. Red and blue arrows indicate the peak positions at $\pm|\Delta_T + E_{YSR}|$ for the interacting impurities A and B with $E_{YSR} = 0.47 \pm 0.01$ meV.

For a given purity (99.9%) of commercially available La pieces, common magnetic and metallic contaminants are Fe, Nd, Ni, and so on (listed in descending concentration)[35]. The magnetic contaminants were co-evaporated unavoidably during the deposition of La, and they are mostly distributed at the subsurface layers of La after post annealing of the sample [Supplementary Note 2 and Supplementary Fig. 2]. The surface cleanliness and the thickness of La films were checked by STM after transferring the sample into the cryostat. A La-coated PtIr tip was used for STM/STS measurements to improve the energy resolution in the spectra at a given experimental temperature via the superconducting tip[14,15,17,21]. To fabricate the La-coated tip in situ, a mechanically polished PtIr tip was intentionally indented into the clean La film. The superconducting La–La junctions were confirmed by observing the Josephson tunneling current at $V = 0.0$ mV as well as the spectral signature of the multiple Andreev reflections.

**STM/STS measurements**. All the experiments were performed in a low-temperature STM system (USM-1300S, Unisoku, Japan) operating at $T = 1.65$ K under ultrahigh vacuum conditions. Tunneling spectra were obtained by measuring the differential tunneling conductance ($dI/dV$) using a standard lock-in technique with a modulation voltage of 30 $\mu V_{rms}$ and a frequency of 1075 Hz with opened feedback loop. The bias voltage was applied to the sample, and the tunneling current was measured through the tip using a commercially available controller (Nanonis, SPECS).

**Electronic structure calculations**. The ab initio calculations were performed in the dhcp structure of La, using the lattice constants of $a_0 = 3.772$ Å and $c_0 = 12.144$ Å. As a first step, slab calculations were performed using the Vienna Ab initio Simulation Package (VASP)[36–38] to determine the relaxation of the top atomic layer on the La(0001) surface. The slab consisted of 8 atomic layers and an about 24-Å-thick vacuum layer to avoid interactions between the periodic images of the supercell. The potential was parameterized based on the PBE method[39], and a $15 \times 15 \times 1$ Monkhorst–Pack mesh was used to sample the Brillouin zone. It was found that the outermost layer relaxes inward by about 5.8% of the interlayer distance $c_0/4$.

Further calculations were performed based on the fully relativistic screened Korringa–Kohn–Rostoker (SKKR) Green's function formalism[40], using the geometry obtained above. After determining the potential of bulk La self-consistently, surface calculations along the [0001] stacking direction were performed for a system consisting of 12 atomic layers of La and 4 atomic layers of vacuum, embedded between semi-infinite bulk La and semi-infinite vacuum. The exchange-correlation potential was parameterized using the method of Vosko, Wilk, and Nusair[41], the atomic sphere approximation was used with an angular momentum cutoff of $l_{max} = 3$, and 24 **k** points were considered in the irreducible part of the Brillouin zone. The LDA + U method[42], as implemented in the SKKR code[43], was used to shift the empty $4f$ bands 5–6 eV above the Fermi level, in agreement with

their experimentally determined position[44]. This was achieved by setting the parameter values to $U = 8.1$ eV and $J = 0.6$ eV, which were reported in ref. [22].

Based on these calculations, the BSF was determined, defined as[45,46]

$$A_l^B\left(E, \mathbf{k}_{\parallel}\right) = -\frac{1}{\pi}\mathrm{ImTr}\int G_{ll}^+\left(E, \mathbf{k}_{\parallel}, \mathbf{r}\right)\mathrm{d}^3\mathbf{r}, \qquad (1)$$

where $E$ is the energy, $\mathbf{k}_{\parallel}$ is the in-plane wave vector, $l$ is the layer index, $G^+$ is the retarded Green's function, and integration is performed over the atomic sphere. As illustrated in Fig. 2b, surface states show up as sharp peaks if $A_l^B(E, \mathbf{k}_{\parallel})$ is plotted as a function of energy at a fixed $\mathbf{k}_{\parallel}$ vector, which enables the determination of a quasiparticle spectrum $E(\mathbf{k}_{\parallel})$. The width of these peaks is determined by the small imaginary part of the energy introduced for numerical stability, which was set to 1 mRy during the calculations. For comparison, bulk properties are illustrated in Supplementary Fig. 6.

**Calculation of the YSR states**. The system was described by a single-band tight-binding model on a two-dimensional triangular lattice

$$H = \sum_{\mathbf{k}} \xi_{\mathbf{k}}\left(c_{\mathbf{k}\uparrow}^{\dagger}c_{\mathbf{k}\uparrow} + c_{\mathbf{k}\downarrow}^{\dagger}c_{\mathbf{k}\downarrow}\right) + \sum_{\mathbf{k}} \Delta\left(c_{\mathbf{k}\uparrow}^{\dagger}c_{-\mathbf{k}\downarrow}^{\dagger} + c_{-\mathbf{k}\downarrow}c_{\mathbf{k}\uparrow}\right)$$
$$- \sum_{\mathbf{k},\mathbf{k}'} \frac{JS}{2N}\left(c_{\mathbf{k}\uparrow}^{\dagger}c_{\mathbf{k}'\uparrow} - c_{\mathbf{k}\downarrow}^{\dagger}c_{\mathbf{k}'\downarrow}\right) + \sum_{\mathbf{k},\mathbf{k}'} \frac{K}{N}\left(c_{\mathbf{k}\uparrow}^{\dagger}c_{\mathbf{k}'\uparrow} + c_{\mathbf{k}\downarrow}^{\dagger}c_{\mathbf{k}'\downarrow}\right), \qquad (2)$$

where $c_{\mathbf{k}\sigma}, \sigma \in \{\uparrow, \downarrow\}$ is a fermionic annihilation operator. $\xi_{\mathbf{k}}$ is the quasiparticle spectrum in the normal state, which was determined from the ab initio calculations; see Supplementary Table 1 for details. $\Delta = 0.95$ meV is the superconducting order parameter. The magnetic impurity in the system is represented by a classical spin oriented along the $z$ direction, with $\frac{JS}{2}$ magnetic and $K$ nonmagnetic scattering potentials, and $N = 1024 \times 1024$ denotes the number of sites. The impurity creates a single pair of quasiparticle eigenstates inside the gap, the energy and the wave function of which was determined, and the corresponding electron density plotted in Fig. 3c, see Supplementary Note 6 for details of the calculation procedure. The values $\frac{JS}{2} = 0.735$ eV and $K = 0$ eV were selected to reproduce the experimentally observed energy of the YSR state, $E_{YSR} = \pm 0.55$ meV. For details on the quasiparticle spectrum used for the circular and hexagonal FS in Fig. 3a, b see Supplementary Note 5 and Supplementary Fig. 7.

## Data availability
The authors declare that the data supporting the findings of this study are available within the paper and its Supplementary Information files.

## Code availability
The computational codes for the tight-binding model calculations of the band dispersion and the spatially resolved YSR-bound states are available from the authors upon reasonable request.

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

## Acknowledgements

We would like to thank J. Wiebe, Y. Nagai, D.K. Morr, L. Szunyogh, and A. Deák for the helpful discussions. This work was supported by the European Research Council via project No. 786020 (ERC Advanced Grant ADMIRE), the Alexander von Humboldt Foundation, and by the National Research, Development, and Innovation Office of Hungary under project No. K131938.

## Author contributions

H.K. and R.W. conceived and designed the experiments. H.K and D.S. carried out the STM/S experiments. H.K. analyzed the data. L.R. and E.S. performed the first-principles calculations and provided their analysis. H.K. and L.R. did the atomistic model calculations. R.W. supervised the project. H.K., L.R., and R.W. wrote the paper. All authors discussed the results and contributed to the paper.

## Funding

## Competing interests

The authors declare no competing interests.
