## [Peer Review File · Nature Communications]

REVIEWER COMMENTS

Reviewer #1 (Remarks to the Author):

Kim and coauthors reported here on the extension of the wave function of impurity bound states on a superconducting La film. The manuscript clearly shows that YSR bound states presumably caused by magnetic impurities (of unknown nature) can be excited from extremely large distance apart, depicting a beautiful physical platform for studying superconducting quasiparticle excitation. The main point of novelty here is the larger extension of the YSR excitation beams in La, which are observed for more than 30 nm apart, more than 3 times larger than in NbSe₂. This is remarkable indeed and opens the door to the study of many new properties of these bound states in exotic scenarios such artificial atomic structures, for which the authors did a very basic demonstration with the measurement of the interaction between two YSR states close one another. The paper is also complemented with interesting model calculations which are mostly presented as supplementary information. In general, I congratulate the authors for this finding.

I must admit that, although I enjoyed reading the manuscript a lot, I found it overall a bit descriptive and many statements are rather qualitative. The reported long extension is evident from the provided results and its correspondence with the anisotropic surface band of La is well demonstrated. However, the paper fails to outline the new aspects with respect to previous papers, like for example, the one by Ménard and coworkers. So, although the paper is of sufficient interest for the community accessing to the submitted journal, I would suggest working out some of the most descriptive parts and discussing a bit on physical mechanism. I mention some points that could help to improve the novelty and the impact of this paper, but I am sure the authors will find others:

1- I suspect that the long extension is also connected to the lifetime of the YSR quasiparticle. Can the authors make some estimates?

2-Can they compare with non-superconducting tips? Are the beams' extension similar?

3- is the long range due to the focusing effect or to the lower dimensionality of the band? Can it be distinguished? the authors could provide a comment on this, especially after the very nice simulation in supplementary material figure 4, which would be an advance.

4- The magnetic coupling between impurities (Fig. 4) is the most superficial aspect. Can the authors provide more data here that would eventually relate the distance and orientation of the impurities to a type of coupling? I am aware that a proper dataset this might difficult to obtain for subsurface impurities.

5-How precise are the YSR peak values in a set of impurities? I suspect some dispersion around some nominal values may exist, simply due to the proximity of several impurities. Is the statistical dispersion smaller than the 90 μeV observed for the coupled one? This statistic is necessary to enforce the value of the reported interaction.

6- In page 3, the authors relate the different bound states (YSR1 to YSR4) to different element and depth. While the depth seems obvious given that they are subsurface impurities, the different element needs to be demonstrated. They miss to provide a comparison of peak values (or comparative spectra as Supplementary Information) to substantiate this rather descriptive statement.

The authors certainly did a nice scholar demonstration of the focusing effect. I hope that this can also be used to unravel some of the physics behind.

Besides these points, I found a few issues in the manuscript that the authors should be aware of:

- a) The word "Gigant" in the title is unjustified. The starting of the title "Long-range" already describes what is reported.
- b) The term "Friedel oscillations" describes oscillations in the electron density (charge/volume) around an impurity potential and is strictly defined at the Fermi energy. Although this term is frequently used to describe any type of oscillation caused by scattering potentials, perhaps the authors should use some other terminology that more appropriately expresses standing waves in the density of states (# states/volume/energy).
- c) the term "perpendicular to the beams" is misleading. The directions Q1 in fig. 1e are not perpendicular to the beams.
- d) The real space dI/dV image at 3 meV (Sup. Fig. 3(b)) should be shown in the main manuscript for reference. I find still instructive to show the data from where the FFT map was measured.
- e) In Page 5, the sentence: "... from intra-band scattering of electrons located in the surface band" is not accurate. At 3 meV there are no electrons located in the band because it is empty. The scattering element is a quasiparticle state excited by the tunneling electron.
- f) In page 5: the word "insensitivity" of bulk bands to scattering is not appropriate. Note that in the next sentence a reference to the case of Pb(111) (ref. 19) is made, where the band responsible of the YSR oscillations is a bulk band.
- g) One of the conclusions is that the "effective reduction in dimensionality ..." facilitates "...the long-range extension of the bound states". This conclusion was not discussed neither shown in the paper (as noted in my comment 3) above).

A couple of typos:

Supplementary Information, Line 4 in page 4 has a squared symbol

Supplementary Information, page 5, one before the last sentence, "hole part" should replace "particle part".

Reviewer #2 (Remarks to the Author):

This manuscript reports on YSR states created by magnetic impurities in a 3D La superconductor. The authors mainly discuss on the spatial distribution and the symmetry of the these states. The authors showed that the long-range spatial extension of the YSR state near the subsurface layers is due to the existence of the quasi-2d surface state existed on the (0001) surface of the bulk La. This is in contrast to conventional YSR states in a 3D superconductor, which decay much faster. Moreover, it was demonstrated that due to the hexagonal shape of the Fermi surface, the YSR state has a six-fold symmetry with a star shape. The presented experimental results are in high quality and the explanations with theoretical modeling are overall convincing. Thus, I would like to recommend this work to be published in Nature Communications.

Reviewer #3 (Remarks to the Author):

The authors report that magnetic impurities embedded below a superconducting La(0001) surface interact via quasiparticles extending to very large distances by low temperature STM/STS. They find the corresponding anisotropic and giant oscillations in the local density of states and give a reasonable explanation. This is an interesting work, I would like to recommend it publish in NC, however, several questions should be addressed.

1. How deep are those impurities normally?
2. The authors attribute the large the extension to the anisotropic Fermi surface formed by the surface band on La(0001). How can the magnetic impurities interact with the surface band of La?
3. Have they tried to put some Fe impurities on the surface of La(0001)? If so, any difference?

Response to Reviews

We thank all Reviewers for the careful reading of our manuscript and providing their comments, as well as for the positive evaluations. In the following, we provide point-by-point replies to their comments. We revised the manuscript based on all Reviewers' concerns and suggestions in order to improve the discussion on the physical mechanisms of the long-range extension of the YSR bound states on La(0001), and highlighted the changes in the text in yellow. We hope that our revised version of the manuscript can soon be published in Nature Communications.

Comments of Reviewer 1:

Reviewer #1 (Remarks to the Author):

Kim and coauthors reported here on the extension of the wave function of impurity bound states on a superconducting La film. The manuscript clearly shows that YSR bound states presumably caused by magnetic impurities (of unknown nature) can be excited from extremely large distance apart, depicting a beautiful physical platform for studying superconducting quasiparticle excitation. The main point of novelty here is the larger extension of the YSR excitation beams in La, which are observed for more than 30 nm apart, more than 3 times larger than in NbSe₂. This is remarkable indeed and opens the door to the study of many new properties of these bound states in exotic scenarios such artificial atomic structures, for which the authors did a very basic demonstration with the measurement of the interaction between two YSR states close one another. The paper is also complemented with interesting model calculations which are mostly presented as supplementary information. In general, I congratulate the authors for this finding.

I must admit that, although I enjoyed reading the manuscript a lot, I found it overall a bit descriptive and many statements are rather qualitative. The reported long extension is evident from the provided results and its correspondence with the anisotropic surface band of La is well demonstrated. However, the paper fails to outline the new aspects with respect to previous papers, like for example, the one by Ménard and coworkers. So, although the paper is of sufficient interest for the community accessing to the submitted journal, I would suggest working out some of the most descriptive parts and discussing a bit on physical mechanism. I mention some points that could help to improve the novelty and the impact of this paper, but I am sure the authors will find others:

Our response:

→ We appreciate the Reviewer's constructive comments and his/her positive evaluation of our manuscript. Regarding the Reviewer's concerns about the description of the physical mechanism, the main novelty of our studies is that the long-range extension of the YSR bound states may not only be observed in layered materials as suggested by Ménard et al. (Ref. (20)), but in bulk superconductors as well, provided that there is a pronounced surface band involved in the quasiparticle excitations at the magnetic impurities. Furthermore, our work provides a clear experimental demonstration and theoretical confirmation of the quasiparticle focusing effect through the Fermi surface leading to an exceptionally long-range extension of the YSR states, which has not been reported previously. We revised the manuscript to improve the relevant parts emphasizing the novelty and the impact of our work based on the Reviewer's fruitful suggestions. In the following, we provide point-by-point replies to his/her comments.

1- I suspect that the long extension is also connected to the lifetime of the YSR quasiparticle. Can the authors make some estimates?

→ As the Reviewer pointed out, the spatial extension of the electronic excitation states may presumably be linked to the lifetime of the excited quasiparticles on the surface. We agree that studying this relationship between the spatial distribution of the YSR states and their lifetime can be interesting.

We can estimate the lifetime of the YSR state for the subsurface magnetic impurities based on the analysis of the spectral shape of the YSR peak. This provides an upper bound for the lifetime broadening of the YSR states (due to instrumental limitations and finite temperature effects). As shown in Fig. R1, we find a spectral width of 0.19 meV for the YSR1 impurity. Based on the uncertainty relation, this width relates to ~ 21 ps as an upper limit of the intrinsic lifetime of this YSR state.

On the other hand, referring to the model of the spatial extent of the YSR bound states in the Supplementary Note 5, the term describing the exponential decay away from the impurity is connected to the phase relaxation length L_ϕ of the quasiparticle, which provides an estimate for the lifetime of the quasiparticle [PRL 82, 4516 (1999)]. With the parameters of $\Delta=1$ meV and $E_{\text{YSR}}=0.55$ meV, we can estimate the lifetime of the YSR quasiparticle to 0.8 ps, which can be considered as a lower bound of the lifetime.

However, we note that this analysis is not sufficient to conclude about the relationship between the lifetime of YSR states and their spatial extensions. For example, Etzkorn et al. [arXiv. 1807.00646 (2018)] reported on the spatial extension of YSR states for a magnetic impurity on superconducting V(100). Although the spectral width of the YSR states for the magnetic CuPc molecule on V(100) was several tens of μeV , which is much smaller than the value we obtained for our magnetic impurities in La(0001), the spatial extension of the YSR states in their case was only up to a few nanometers, which is much shorter than in our case (~ 30 nm). This example clearly shows that the expectation that a longer lifetime should correspond to a larger spatial extension of the quasiparticle states is not fulfilled.

In conclusion, although the relationship between the lifetime of the YSR bound states and their spatial extension is an interesting subject, we would like to leave this issue for more systematic investigations in the future based on possible time-resolved spectroscopic measurements together with rigorous theoretical model considerations focusing on the dynamics of the YSR states.

Fig. R1: Analysis of the spectral line width for the data presented in Fig. 1(d) of the main text. The experimental curves obtained at (a) the YSR1 impurity and (b) the impurity-free La(0001) surface can be compared to the Lorentzian fits (blue and red solid lines) at (a) $V=-(E_{\text{YSR}}+\Delta_{\text{Tip}})/e$ for the YSR peak and (b) $V=+(\Delta_{\text{Surf}}+\Delta_{\text{Tip}})/e$ for the superconducting coherence peak.

2-Can they compare with non-superconducting tips? Are the beams' extension similar?

→ We appreciate the Reviewer's suggestion. First of all, as the Reviewer probably knows from previous STM/STS studies on superconducting substrates, we would like to emphasize that for investigations of superconductors by STM/STS the use of superconducting probe tips, as in our present study, is a well-established procedure in order to improve the energy resolution of tunneling spectroscopic measurements beyond the limitations set by thermal broadening. [see e.g.: S.-H. Ji *et al.*, *Phys. Rev. Lett.* **100**, 226801 (2008); K. J. Franke *et al.*, *Science* **332**, 940 (2011); M. Ruby *et al.*, *Phys. Rev. Lett.* **115**, 197204 (2015); G. Ménard *et al.*, *Nature Commun.* **8**, 2040 (2017); D.-J. Choi *et al.*, *Phys. Rev. Lett.* **120**, 167001 (2018); H. Kim *et al.*, *Sci. Adv.* **4**, eaar5251 (2018)]. There have been no indications of artifacts in the spatial distribution of the electronic states on superconducting surfaces due to the use of superconducting tips in scanning tunneling spectroscopic measurements, except for the Josephson tunneling regime providing information on transport properties specific to superconductor-superconductor junctions. Unfortunately, tunneling spectroscopic maps obtained with a non-superconducting STM tip for the thick La films (> 50 nm) are not available. Instead, Figure R2 presents spatially-resolved dI/dV maps for a thinner La film (~ 12 nm) hosting the same kind of magnetic impurities using a non-superconducting PtIr tip. Although the thickness of the La film is different in this case and the concentration of the magnetic impurities is higher, one can clearly realize the similar signature of the anisotropic long-range extension of the YSR states. Therefore, we can conclude that STM/STS results obtained with a non-superconducting tip would not modify the discussions and the main conclusions in the present manuscript which focuses on experimental results as obtained with superconducting probe tips.

Fig. R2: Spatially-resolved LDOS maps for subsurface YSR impurities below the surface of the thinner La film using a non-superconducting PtIr tip at $T=1.7$ K. (a) STM topography image at $V=+0.025$ mV (b) the simultaneously obtained dI/dV map at $V=+0.025$ mV. (c) the dI/dV map at $V=-0.025$ mV. Scale bar in (a): 10 nm. The tunneling current was stabilized at $I_T=0.2$ nA.

3- is the long range due to the focusing effect or to the lower dimensionality of the band? Can it be distinguished? the authors could provide a comment on this, especially after the very nice simulation in supplementary material figure 4, which would be an advance.

- In the manuscript, we have considered both the dimensionality of the system and the quasiparticle focusing effect through the Fermi surface as key mechanisms for the long-range extension of YSR bound states. We have come to the conclusion that the focusing due to the anisotropic Fermi surface and due to the reduced dimensionality cannot be rigorously distinguished, because they are essentially different manifestations of the same mechanism. For example, an isotropic two-dimensional Fermi surface (circle) corresponds to a cylindrical three-dimensional Fermi surface, which is strongly anisotropic since the quasiparticles cannot propagate along the axis of the cylinder. Incidentally, the innermost sheet of the Fermi surface of bulk La is almost perfectly cylindrical according to our calculations, see Supplementary Fig. 6(b) and (c). Conversely, the contribution of the anisotropic Fermi surface may be understood as an effectively reduced dimensionality, leading to a reduced absolute value of the decay exponent $|b|$ in Supplementary Fig. 5. However, we find that a circular two-dimensional or cylindrical three-dimensional Fermi surface could not fully account for the long-range extension, since they would lead to a characteristic $1/r$ decay that is faster than the spatial decay observed in the experiments ($\sim 1/r^{0.92}$). To elucidate all possible contributions to the long-range extension of the YSR states, it is necessary to consider more realistic model descriptions of the scattering processes in superconductors, taking into account information about the rigorous band structure of the system as well as the exact location of the magnetic impurity relative to the host lattice.
- We added the corresponding discussion in the revised manuscript on page 6, in the second paragraph starting on line 6.

4- The magnetic coupling between impurities (Fig. 4) is the most superficial aspect. Can the authors provide more data here that would eventually relate the distance and orientation of the impurities to a type of coupling? I am aware that a proper dataset this might difficult to obtain for subsurface impurities.

- We appreciate this suggestion of the Reviewer. The best way to demonstrate the mutual interactions between YSR impurities is given by a reliable control of their distance and relative orientation using STM-based atom-manipulation, which we previously demonstrated for the system of magnetic Fe atoms on a superconducting Re(0001) surface [H. Kim et al., Sci. Adv. 4, eaar5251 (2018)]. However, as the Reviewer has already noted, it is very difficult to control the relative positions of and investigate the mutual interactions between subsurface YSR impurities, which are typically not accessible by single-atom manipulation techniques. This is why we had to rely on the distribution of the intrinsic magnetic impurities formed during the deposition of the La film. In certain cases, as shown in Fig. 4, this leads to YSR impurities facing each other, thereby providing a means to study their mutual interaction in terms of the energy shift of the YSR peaks, being identical for both impurities.

5-How precise are the YSR peak values in a set of impurities? I suspect some dispersion around some nominal values may exist, simply due to the proximity of several impurities. Is the statistical dispersion smaller than the 90 micro-eV observed for the coupled one? This statistic is necessary to enforce the value of the reported interaction.

- As the Reviewer pointed out, the energy shift due to the mutual interactions between YSR impurities should be compared to the distribution of the YSR bound state energy (E_{YSR}) for different YSR impurities. We have checked E_{YSR} of numerous YSR1 impurities, resulting in a

variation of E_{YSR} in tunneling spectra (an example is shown in Fig. R3). We found quite consistent E_{YSR} values for the isolated YSR1 impurities of the La film within an error of $\pm(30-40)$ μeV around the value of $E_{\text{YSR1}}=0.55$ meV as reported in the main text (see also the spectra for the isolated YSR1 impurities in Fig. 4(b)). Therefore, we can conclude that the 90 μeV shift of the YSR peak results from the mutual interactions between these impurities. Nevertheless, we would like to note that the exact mechanism of the spectral variation for YSR pairs should be addressed by rigorous theoretical calculations based on the variety of geometrical configurations of the interstitial magnetic impurities within the La host lattice.

Fig. R3: Tunneling spectra for the isolated YSR1 impurities obtained on different areas of the (0001)-oriented surface of the La film using a La-coated superconducting tip, whose superconducting gap (Δ_T) is 0.89 meV. The extracted YSR binding energy (E_{YSR}) for each spectrum is displayed in the middle of the plot. Experimental temperature $T = 1.65$ K. The tunneling current was stabilized at $I_T=1.0$ nA.

6- In page 3, the authors relate the different bond states (YSR1 to YSR4) to different element and depth. While the depth seems obvious given that they are subsurface impurities, the different element needs to be demonstrated. They miss to provide a comparison of peak values (or comparative spectra as Supplementary Information) to substantiate this rather descriptive statement.

→ The YSR binding energy, E_{YSR} , is regarded as the characteristic parameter for the YSR impurity being dependent on the magnetic moment of the magnetic impurity as well as the magnetic exchange coupling to the superconductor. However, the magnetic moments of the YSR impurities are typically sensitive to the local environment of the impurity within the host lattice (see, e.g., Ref. (18): L. Schneider et al., npj Quantum Materials 4, 42 (2019), for different adsorption sites of the same magnetic impurity). Therefore, it is impossible to specify the magnetic elements based on their spectral signatures in the tunneling spectra.

In this study, the YSR impurities were introduced by the film deposition of La, whose purity was 99.9+%. The common metallic impurities in the La material, used as evaporant, are Fe, Nd, and Ni. Therefore, it is reasonable to assume that the YSR impurities originate from the presence of different magnetic elements. However, as discussed above, it is not possible to unambiguously conclude that the different bound states can be attributed to different elements, and therefore we have rephrased the corresponding sentence in the manuscript as follows:

- Page 3, line 22: “The various shapes of the modulations (YSR1, YSR2, YSR3 and YSR4 in Figs. 1(a) and (b)) presumably originate from the various depths of the magnetic impurities [see Supplementary Note 2]. However, it is also possible that different magnetic elements being present as natural impurities even in high-purity La material may contribute to the different spatial and spectroscopic signatures of the observed YSR impurities.”
- We added this discussion to the Supplementary Information as the new Supplementary Note 3 with Supplementary Figure 3.

Fig. R4: Tunneling spectra obtained at the center of YSR1, YSR2, YSR3 and YSR4 impurities at $T=1.65$ K. All curves are vertically shifted for clarity. All spectra have been measured with a La-coated superconducting tip whose superconducting gap is 0.6 meV as marked by green dotted lines. For reference, a typical spectrum (gray curve) obtained at a far distance away from all YSR impurities on the pure La(0001) film is presented at the bottom. The stabilization tunneling conditions are $I_T=1.0$ nA and $V_S=3.0$ mV.

The authors certainly did a nice scholar demonstration of the focusing effect. I hope that this can also be used to unravel some of the physics behind.

- We appreciate the Reviewer’s positive statement about our demonstration of the focusing effect of YSR states. We indeed demonstrated and emphasized the importance of the Fermi surface and its crucial role for the focusing effect of the YSR bound states resulting in an exceptional long-range extension of the YSR bound states. This effect is new and has not been considered before. It goes beyond the argument of dimensionality effects as discussed previously in the context of layered superconductors. As the Reviewer stated, we also believe that our work will stimulate the consideration of the Fermi surface of materials and the related quasiparticle focusing effect for various quasiparticle-based phenomena, including vortex bound states and even Majorana bound states in (topological) superconducting systems.

Besides these points, I found a few issues in the manuscript that the authors should be aware of:

a) The word “Gigant” in the title is unjustified. The starting of the title “Long-range” already describes what is reported.

→ We adopted the Reviewer’s suggestion and removed the word “giant” from the title.

b) The term “Friedel oscillations” describes oscillations in the electron density (charge/volume) around an impurity potential and is strictly defined at the Fermi energy. Although this term is frequently used to describe any type of oscillation caused by scattering potentials, perhaps the authors should use some other terminology that more appropriately expresses standing waves in the density of states (# states/volume/energy).

→ We appreciate the Reviewer’s comment and chose to rephrase the parts where Friedel oscillations were mentioned in the revised manuscript.

→ Page 2, line 10: “The FS is defined in the reciprocal space of crystals, but its influence is also manifest in real space in the vicinity of impurities via screening of the impurity potential resulting in a modulation of the electron density, also known as Friedel oscillations”.

→ Page 4, line 24: “The atomic defects close to the surface create oscillatory quasiparticle interference (QPI) patterns in the measured dI/dV maps [See Supplementary Note 4]”

→ Page 5, line 1: “Since both the quasiparticle scattering at the Fermi energy and the YSR states find their origin in the FS, it is reasonable to consider the analogies between them.”

→ Supplementary Note 4, Page 3, line 17: “The standing-wave patterns produced by the electron scattering around the surface defects provide information about the dispersion of the surface band structure via energy-dependent quasiparticle interference (QPI) imaging followed by Fourier transformation of the real-space data.”

c) the term “perpendicular to the beams” is misleading. The directions Q1 in fig. 1e are not perpendicular to the beams.

→ We thank the Reviewer for the comment and realized that using the term “perpendicular to the beams” may be misleading because the direction Q1 is perpendicular to one beam, but forms an angle of 30° with the other two beams. Therefore, we replaced it with “along the direction halfway between the beams”

→ Page 4, line 1: “and up to ~10 nm distance along the directions halfway between the beams,”

→ Page 4, line 12: “The period of the oscillations, corresponding to half of the Fermi wavelength ($\lambda_F/2$) [Supplementary Note 5], is found to be ~0.99 nm along the beams (Q2 direction in Fig. 1(e)) and ~0.90 nm along the direction halfway between the beams (Q1 direction), respectively, for both positive and negative bias voltages.”

d) The real space dI/dV image at 3 meV (Sup. Fig. 3(b)) should be shown in the main manuscript for reference. I find still instructive to show the data from where the FFT map was measured.

→ We thank the Reviewer for the suggestion. While we agree that the constant-current image of Supplementary Fig. 4(b) is instructive to experts, we deem that it is not essential for understanding the results in the main manuscript concentrating on the Fermi surface focusing. Therefore, we prefer to leave this figure in the Supplementary Information.

e) In Page 5, the sentence: "... from intra-band scattering of electrons located in the surface band" is not accurate. At 3 meV there are no electrons located in the band because it is empty. The scattering element is a quasiparticle state excited by the tunneling electron.

→ Following the recommendation of the Reviewer, we rephrased the sentence as:

→ *Page 5, line 14: "it can be concluded that the hexagonal QPI pattern in Fig. 2(c) arises from intra-band scattering of quasiparticles originating from the surface band (inner hexagon in Fig. 2(d))."*

f) In page 5: the word "insensitivity" of bulk bands to scattering is not appropriate. Note that in the next sentence a reference to the case of Pb(111) (ref. 19) is made, where the band responsible of the YSR oscillations is a bulk band.

→ We followed the Reviewer's suggestion and rephrased that sentence in the revised version of our manuscript.

→ *Page 5, line 18: "This implies that the surface band of La(0001) plays a major role for the quasiparticle scattering at the surface while the contribution of the bulk band is weak or lacking."*

g) One of the conclusions is that the "effective reduction in dimensionality ..." facilitates "...the long-range extension of the bound states". This conclusion was not discussed neither shown in the paper (as noted in my comment 3) above).

→ In order to clarify the meaning of "effective reduction of dimensionality", we would like to point out that the power-law decay of the LDOS in YSR states follows r^{1-d} , as discussed in Supplementary Note 5 and shown earlier for two- and three-dimensional systems by Ménard et al. (Ref. (20)). As written on page 5 of our manuscript, "For the hexagonal Fermi surface in Fig. 3(b), the LDOS along the beam directions perpendicular to the flat sides of the FS decays slower than $1/r$, effectively reducing the dimensionality due to the strong focusing effect", i.e. the observation of an exponent closer to zero may be interpreted as a reduced (fractional) dimension based on the above power law. In our reply to comment 3, we tried to clarify that the reduced dimensionality and the anisotropy of the Fermi surface lead to the same physical consequences. To remove the remaining ambiguity, we chose to rephrase the sentence in the conclusion as:

→ *Page 7, line 5: "The results confirm that the slow power-law decay and long-range extension of the YSR bound states, which may be interpreted as an effective reduction in dimensionality, can be observed not only in layered systems, but also in elemental bulk superconductors exhibiting distorted FSs with quasi-2D bands."*

A couple of typos:

Supplementary Information, Line 4 in page 4 has a squared symbol

- We expect that this error occurred during the conversion process of the file type in the submission server. Although we could not find the part in the manuscript file as pointed out by the Reviewer, we will carefully check all symbols after converting the file.

Supplementary Information, page 5, one before the last sentence, “hole part” should replace “particle part”.

- In order to clarify this sentence, we revised it as follows:
- *Supplementary Note 6, page 6, the last sentence: “Due to the particle-hole symmetry of the Hamiltonian, the hole part $v_{-k\downarrow q}$ at ε_q is equal to the particle part of the YSR state observed at the opposite bias voltage $-\varepsilon_q$.”*

Reviewer #2 (Remarks to the Author):

This manuscript reports on YSR states created by magnetic impurities in a 3D La superconductor. The authors mainly discuss on the spatial distribution and the symmetry of these states. The authors showed that the long-range spatial extension of the YSR state near the subsurface layers is due to the existence of the quasi-2d surface state existed on the (0001) surface of the bulk La. This is in contrast to conventional YSR states in a 3D superconductor, which decay much faster. Moreover, it was demonstrated that due to the hexagonal shape of the Fermi surface, the YSR state has a six-fold symmetry with a star shape. The presented experimental results are in high quality and the explanations with theoretical modeling are overall convincing. Thus, I would like to recommend this work to be published in Nature Communications.

Our response:

- We appreciate the Reviewer’s positive evaluation of our manuscript and his/her recommendation for its publication in Nature Communications.

Reviewer #3 (Remarks to the Author):

The authors report that magnetic impurities embedded below a superconducting La(0001) surface interact via quasiparticles extending to very large distances by low temperature STM/STS. They find the corresponding anisotropic and giant oscillations in the local density of states and give a reasonable explanation. This is an interesting work, I would like to recommend it publish in NC, however, several questions should be addressed.

- We would like to thank the Reviewer for his/her constructive comments, for the positive evaluation, and for recommending our manuscript for publication. In the following, we provide point-by-point replies to the Reviewer’s comments.

1. How deep are those impurities normally?

→ In the present study, magnetic impurities have been co-deposited *in situ* with La onto the Re substrate during the electron beam evaporation of La (purity: 99.9+%). Therefore, we expect that the naturally occurring magnetic impurities (primarily Fe, Nd, Ni) are homogeneously distributed within the La film. However, since the STM/STS technique is surface-sensitive and the La(0001) film exhibits a quasi-2D electronic structure with an almost non-dispersive electronic band along the (0001) direction, we expect that the observed YSR bound states in this study mostly originate from the magnetic impurities residing within a few layers relative to the top surface plane.

To identify at least the depth of the YSR1 impurities near the surface, which we have primarily focused on in our analysis, we intentionally removed the atomic layers of La by applying a local voltage pulse of 10 V for 50 ms at the YSR1 impurity site, as shown in Fig. R5. After removing the first atomic layer of La, the signature of the YSR impurity in the dI/dV image has disappeared. This clearly indicates that the YSR1 impurities are located just below the first atomic layer of La.

Fig. R5: Experimental determination of the location of the YSR1 impurity. (a) STM topography image of the isolated YSR1 impurity. (b) Simultaneously obtained dI/dV map at $E=E_{\text{YSR}}$ showing the anisotropic long-range extension of the YSR bound state as marked by a white arrow. (c) STM topography image and (d) simultaneously obtained dI/dV map after applying a voltage pulse of 10 V for 50 ms above the YSR1 impurity. By this procedure, we could locally remove one atomic layer of La resulting in the formation of the artificial vacancy island as marked by a black arrow. The LDOS modulation in (b) has disappeared because the YSR1 impurity has been removed together with the La layer by the voltage pulse. Experimental temperature $T = 1.65$ K. The tunneling current was stabilized at $I_T=1.0$ nA.

2. The authors attribute the large the extension to the anisotropic Fermi surface formed by the surface band on La(0001). How can the magnetic impurities interact with the surface band of La?

→ In this study, we have demonstrated that the surface electronic structure of La(0001) is dominated by the surface band which exhibits quasi-2D behavior and that the directions of the spatially-extended YSR bound states coincide with the geometry of the Fermi surface

originating from the surface band. Based on this, it is a natural conclusion that the conduction electrons in the surface band are responsible for the magnetic exchange interaction with the shallow subsurface magnetic impurities resulting in the formation of the YSR bound states. Furthermore, as demonstrated in the reply to the previous question, the YSR1-type impurity is located below the first atomic layer of the La film, where the LDOS of the electrons in the surface band is high, see Fig. 2(d). Therefore, it is expected that the Cooper pairs formed by electrons in the surface band can scatter off the shallowly buried impurities. For impurities deeply embedded in the bulk where the LDOS of the surface states disappears (see Supplementary Fig. 6(a)), such interactions should not be possible.

3. Have they tried to put some Fe impurities on the surface of La(0001)? If so, any difference?

→ We thank the Reviewer for this question. Yes, we indeed have also tried to deposit Fe atoms on top of the La film. However, due to intermixing between Fe and La, the adsorbed Fe atoms tend to go subsurface and can then hardly be distinguished from the naturally occurring magnetic impurities in the La film. Therefore, these experiments did not provide additional information compared to what is already reported in the manuscript.

REVIEWERS' COMMENTS:

Reviewer #1 (Remarks to the Author):

I thank the authors for the very detailed response to my comments and for having taken into consideration some of my arguments, as well as answered my questions and misunderstandings. I think that the revised version is in perfect shape, and recommend acceptance.

I would just like to point out one issue after the response. The authors state in page 6 of the response that the focusing “..effect is new and has not been considered before”. I am not sure if in the manuscript remains such statement, but if so, I think it should be soften down, because in both papers, by Ménard and by Ruby, the focusing effect caused by some portions of the FS was pointed out as responsible of the observation of extended YSR beams at anisotropic positions. Certainly, the authors here do a nice scholar demonstration with their nice simulations. Otherwise, I find the paper flawless and appropriate for your journal.

Reviewer #3 (Remarks to the Author):

The authors have addressed my concerns and I am satisfied with their answers. Now, I would like to recommend it publish in Nature communications.

Response to Reviews

Comments of Reviewer 1:

Reviewer #1 (Remarks to the Author):

I thank the authors for the very detailed response to my comments and for having taken into consideration some of my arguments, as well as answered my questions and misunderstandings. I think that the revised version is in perfect shape, and recommend acceptance.

I would just like to point out one issue after the response. The authors state in page 6 of the response that the focusing “..effect is new and has not been considered before”. I am not sure if in the manuscript remains such statement, but if so, I think it should be soften down, because in both papers, by Ménard and by Ruby, the focusing effect caused by some portions of the FS was pointed out as responsible of the observation of extended YSR beams at anisotropic positions. Certainly, the authors here do a nice scholar demonstration with their nice simulations.

Otherwise, I find the paper flawless and appropriate for your journal.

Our response:

→ We appreciate again the Reviewer’s constructive comments and his/her positive evaluation of our manuscript during the review process as well as his/her recommendation for publication. The statement the Reviewer pointed out was indeed made in our previous reply, but we confirmed that it is absent from the manuscript. Therefore, it is not necessary to revise the manuscript concerning this statement.

Reviewer #3 (Remarks to the Author):

The authors have addressed my concerns and I am satisfied with their answers. Now, I would like to recommend it publish in Nature communications.

→ We would like to thank the Reviewer for recommending our manuscript for publication.